# Ensemble Learning with Supervised Methods Based on Large-Scale Protein Language Models for Protein Mutation Effects Prediction

**DOI:** 10.3390/ijms242216496

**Published:** 2023-11-18

**Authors:** Yang Qu, Zitong Niu, Qiaojiao Ding, Taowa Zhao, Tong Kong, Bing Bai, Jianwei Ma, Yitian Zhao, Jianping Zheng

**Affiliations:** 1Cixi Biomedical Research Institute, Wenzhou Medical University, Ningbo 315300, China; quyangygs@nimte.ac.cn (Y.Q.); niuzitong@nimte.ac.cn (Z.N.); dingqiaojiao@nimte.ac.cn (Q.D.); zhaotaowa@nimte.ac.cn (T.Z.); 2Cixi Institute of Biomedical Engineering, Ningbo Institute of Materials Technology and Engineering, Chinese Academy of Sciences, Ningbo 315300, China; kongtong@nimte.ac.cn (T.K.); baibing@nimte.ac.cn (B.B.); majianwei@nimte.ac.cn (J.M.)

**Keywords:** machine learning, protein fitness prediction, sequence representation, ensemble learning, protein engineering, deep learning, embeddings

## Abstract

Machine learning has been increasingly utilized in the field of protein engineering, and research directed at predicting the effects of protein mutations has attracted increasing attention. Among them, so far, the best results have been achieved by related methods based on protein language models, which are trained on a large number of unlabeled protein sequences to capture the generally hidden evolutionary rules in protein sequences, and are therefore able to predict their fitness from protein sequences. Although numerous similar models and methods have been successfully employed in practical protein engineering processes, the majority of the studies have been limited to how to construct more complex language models to capture richer protein sequence feature information and utilize this feature information for unsupervised protein fitness prediction. There remains considerable untapped potential in these developed models, such as whether the prediction performance can be further improved by integrating different models to further improve the accuracy of prediction. Furthermore, how to utilize large-scale models for prediction methods of mutational effects on quantifiable properties of proteins due to the nonlinear relationship between protein fitness and the quantification of specific functionalities has yet to be explored thoroughly. In this study, we propose an ensemble learning approach for predicting mutational effects of proteins integrating protein sequence features extracted from multiple large protein language models, as well as evolutionarily coupled features extracted in homologous sequences, while comparing the differences between linear regression and deep learning models in mapping these features to quantifiable functional changes. We tested our approach on a dataset of 17 protein deep mutation scans and indicated that the integrated approach together with linear regression enables the models to have higher prediction accuracy and generalization. Moreover, we further illustrated the reliability of the integrated approach by exploring the differences in the predictive performance of the models across species and protein sequence lengths, as well as by visualizing clustering of ensemble and non-ensemble features.

## 1. Introduction

Proteins play a pivotal role in the fields of life sciences, health, and chemical engineering. They can serve as both integral components of the human body [1] and carriers for transporting various substances [2]. Furthermore, proteins can catalyze biochemical reactions to ensure their smooth progression [3]. However, their natural forms often fail to meet the expectations for advanced applications.

The fields of protein engineering, protein biochemistry, and evolutionary biology have long been interested in the question of where and how to mutate a protein of interest to make the mutated protein more suitable for practical applications. Most of the time, researchers prefer to employ means including directed evolution and rational design to obtain the desired answers and products [4]. Directed evolution is particularly suitable for newly discovered proteins because this approach does not require an in-depth understanding of the sequence, structure, and physicochemical properties of proteins; new and more useful target proteins can be discovered by using only iterative random mutation and artificial selection [5]. However, practically, the advantages of this approach have also become its disadvantages. Among them, the most frequently discussed issues are its unpredictable number of iterations due to random mutations and the high-throughput screening required for manual screening, which are shortcomings hindering the application of directed evolution to many protein species and limiting the process of improvement of some proteins by researchers [6]. Rational design has higher certainty than directed evolution, while the high cost of protein structure analysis has deterred many laboratories, and the specific structure of some proteins cannot be clearly analyzed [7]. Despite the fact that some studies have begun to intervene in the analysis of protein structure by artificial intelligence methods, such as AlphaFold [8], the actual effect of this method has yet to be demonstrated by more experimental studies. Therefore, it is eagerly hoped that newer technological tools will be available to eliminate the existing shortcomings to a certain extent in order to contribute to better protein engineering.

Recently, the field of protein engineering has experienced a significant impact from the advancements in machine learning. To address longstanding challenges, various related methods and tools have been proposed. For instance, AlphaFold [9], a highly precise protein structure prediction model, was developed to overcome the expensive resolution of protein structures, along with its subsequent iterations [10]. Additionally, NetGo [11,12], an annotation tool for protein functionality, and a range of other tools and methods, such as UniRep [13,14], and ESM series [15,16], have been specifically designed for protein fitness prediction, aiming to explore novel or more applicable proteins. Based on the rapid progress in natural language processing (NLP), a subfield of deep learning, researchers have progressively identified commonalities between amino acid sequences and natural language [17,18,19]. This realization has contributed to the continuous development of protein language models, and concurrent investigations into their extended research and practical applications are currently underway. These studies are primarily rooted in the understanding that highly adaptive proteins are preserved during the extended process of natural protein evolution [4,20]. The driving force underlying this phenomenon lies in the concealed evolutionary rules within the vast pool of measured protein sequences. At a certain level, these rules manifest themselves as the syntax and semantics of a distinctive biological language composed of protein sequences. Therefore, protein language models are trained on extensive protein sequences to capture these implicit evolutionary rules and extract semantic information embedded within the protein sequences themselves. Moreover, these models encompass the intricate dependencies among the constituent amino acids that form the proteins. Meanwhile, the emergence of high-throughput sequencing technology [21] has resulted in an exponential expansion of protein sequence databases, including UniProt [22,23] and Pfam [24,25]. Moreover, this expansion provides a solid foundation and conducive circumstances for the training of protein language models.

With the continuous deepening of research on protein language models, an increasing number of protein language models with different architectures and scales have been proposed, and the applications using these models are increasing accordingly. For example, Ali Madani et al. [26] demonstrated that the newly generated artificial protein enzymes produced by ProGen have similar catalytic efficiency to the natural protein enzymes used in specific tasks. Based on fine-tuned protein language models, Surojit Biswas et al. [14] proposed β-lactamase and GFP with higher adaptability compared to the WT (wild type). Suresh Pokharel et al. [27] used protein language models to predict protein succinylation sites and other information. Among them, the study of predicting protein functional characteristics quantitatively based on features extracted by language models directly from protein sequences has attracted particular attention. Different from qualitative predictions categorizing protein sequences into different functional classes, quantitative predictions not only serve as a key component for machine-learning-assisted directed evolution but also provide a clearer exploration of evolutionary relationships between protein sequences, thereby contributing to a deeper understanding of natural selection relationships [28]. The existing studies have shown that models trained on a larger amount of protein sequence data tend to extract richer semantic information due to the highly complex nature of evolutionary rules [29,30]. As a result, they achieve better performance in various downstream tasks. Therefore, currently, researchers are inclined to investigate how to make the models more complex and larger in order to capture the hidden “grammar” and “semantic” information in protein sequences. This “arms race” of large models has been ongoing for some time, from the 85M-parameter Primer S [31] to the current 700 M-parameter Tranception L [32]. Moreover, it is expected that more and larger models will be developed in the future. However, the computational cost of training a large-scale protein language model is extremely high, and many “big models” trained at great cost may not have fully utilized and explored their potential. While different models employ different construction and training methods, their training data generally come from well-known databases such as UniProt [22,23] and Pfam [24,25]. However, the intersection of such training data can probably lead to complementarities in the knowledge learned by different models. The research conducted by Brian L. Hie et al. [33] seems to provide some possibilities from another perspective. They experimentally tested the intersection of reasonable mutations proposed by different existing protein language models and successfully evolved human antibodies. This may indicate that there are differences and commonalities in the features extracted by different protein language models, with certain algorithms being better at capturing one type of information, while others capture another type more comprehensively. Regarding the problem of accurately and efficiently predicting the quantitative functional changes caused by mutations at the sequence level, most protein language models are trained through self-supervision, and they have not fully utilized the measured mutation effects data [26,34,35,36,37]. This also leaves a great deal of room for improvement in prediction accuracy and generalization capability.

In this study, we propose an ensemble learning approach for protein mutation effect prediction tasks. This approach employs pre-trained protein language models of different scales trained on different various databases in order to take advantage of the potential complementarity among the feature information extracted by different models. This approach aims to better simulate the evolutionary relationships between species, capturing the semantic grammar in protein sequences as well as background knowledge in association with structure and stability. Furthermore, we incorporate local evolutionary context into the ensemble representation, inspired by the method of explicitly capturing residue interactions using CCMPred [38] in ECNet [39]. This integration maximizes the utilization of MSAs (multiple sequence alignments) from different protein sequences. Finally, we construct a ridge-based machine learning model employing ensemble representation to map sequences to quantified functionalities. In addition, we explore the visualization clustering performance of different representation methods to show the reliability of the integrated approach. Through benchmark experiments performed on 17 collected DMS (deep mutational scanning) datasets, we find that the ensemble learning approach outperforms existing supervised or unsupervised methods on the majority of the data. Compared to the prediction solely based on individual features, the ensemble learning approach further enhances the accuracy and generalization of model predictions.

## 2. Results

### 2.1. Prediction of Protein Mutation Effects

To demonstrate the utility of our approach, we tested it on S17 (a dataset of 17 deep mutation scans obtained by conditional screening of ProteinGym [32], Section 4.1). Since protein engineering is more concerned with which mutants have higher fitness compared to others in practical applications, in most studies, Spearman rank correlation [39,40] was the main metric considered in our tests. The results were compared with the state-of-the-art supervised and unsupervised methods, including Tranception [32], EVE [41], EVmutation [36], and ECNet [39]. Tranception has developed numerous different versions in its research. In order to better reflect the validity of the ensemble method proposed in this study, we chose to use the most complete version of these methods for comparison; i.e., homologous sequences were retrieved at inference time, and the final results were averaged with the use of two-way inference. This is also the best-performing approach in the study.

The prediction results for this component can be obtained from Figure 1. Specifically, Tranception, EVE, and EVmutation exhibit average Spearman rank correlations of 0.432, 0.473, and 0.426, respectively. By contrast, our method outperforms all the tested models, yielding superior performance. Notably, our approach achieves an average Spearman rank correlation of 0.711 on S17, constituting a substantial improvement of approximately 70%. What is also clearly observable is that our method predicts generally better than ECNet, probably the best existing supervised model, even though it has correlations as high as 0.97 on the data for some specific protein species.

### 2.2. Improved Prediction Performance of Mutation Effects Benefited from the Ensemble of Multiple Global Features

In order to examine the effectiveness of different models and their integration, we conducted an ablation experiment on S17 (Figure 2), assessing the predictive performance of three distinct approaches. These approaches include the use of Tape-transformer [56] or Tranception L as a standalone pre-trained protein language model, as well as the integration of both models. In the supervised inferential portion of the experiment, we uniformly applied a simple ridge regression model to determine whether the ensemble method can favor the mapping of the model from sequence to function.

The disparities between the ensemble and non-ensemble methods are evident in the results. The ensemble method outperformed the non-ensemble method on 15 of the 17 protein datasets, where it was able to improve the correlation up to nearly 0.3. When comparing the test outcomes of the two non-integrated approaches, an intriguing phenomenon emerged. It can be observed that, for certain protein datasets, the prediction performance of the Tape-transformer alone surpassed that of the method solely employing Tranception L by up to 31.59%. However, for other protein datasets, the situation reversed, with the method using only Tranception L exhibiting superior performance by up to 65.07%. This suggests that non-ensemble models appear to be selective about the protein species. Moreover, the ensemble method is in most cases superior to methods that use only one model alone.

### 2.3. Localized Features Further Enrich the Feature Information

Local features in the context of protein sequences generally pertain to specific characteristics derived from shorter contiguous segments exhibiting similarities within a subset of protein sequences [20]. These local features are typically computed with algorithms being applied to the identified subset, emphasizing essential components related to protein structure, function, and interactions. Previous investigations have established a connection between local features and protein functional fitness. For instance, Luo et al. [39] showed the correlation between residue coevolution, computed via CCMPred, and the fitness of double mutants in the human YAP65 WW [57]. This residue coevolution was incorporated into ECNet as a local feature, contributing to improved prediction accuracy. Nonetheless, it remains uncertain whether the integration of the same local feature would still have a positive impact on the inference capabilities of our proposed method. Furthermore, we aim to explore any potential specificity relationship that might exist between the local feature and the selected global feature within our integration approach.

We investigated this aspect through conducting a series of ablation experiments, in which we examined the model’s performance on S17 under six different scenarios (Figure 3), each altering only the input of feature information. Our findings are compelling as they clearly show that methods incorporating “local features” outperform their counterparts that do not exploit such information in all six cases. Averagely, we observed an improvement of approximately 0.1 in Spearman rank correlation. This result indicates that incorporating residue coevolutionary information, computed by CCMPred, as local features can significantly enrich the feature information and enhance the model’s ability to capture the mutation effect values of the target proteins. In addition, the overall enhancement in prediction results suggests that no specific dependency exists between the local and global features. Consequently, our proposed ensemble method effectively benefits from the incorporation of local features, reinforcing the efficacy of our approach.

## 3. Discussion

In the realm of deep learning, various disciplines are advancing towards the development of “large models” [13]. In the field of protein engineering, where the protein sequence space is incredibly vast and biological evolution follows intricate laws, large-scale “protein models” endowed with numerous parameters have proven to be particularly effective for addressing complex problems [20]. However, the high computational cost associated with such models has deterred a large number of laboratories. Therefore, numerous research groups are exploring how to extract more usability from open-source models. Among the challenges encountered in protein engineering, the task of accurately mapping protein sequences to functional quantification of variants represents the largest difficulty [32]. Although many models have demonstrated excellent results in tasks including three-dimensional structure prediction [10], thermal stability prediction [58], and functional annotation [12], the scarcity and integration complexity of measurement data from laboratory experiments, compounded by the diverse functionalities of protein mutants and screening methods used, hinder accurate predictions in the field. Therefore, it becomes crucial to develop prediction methods with higher accuracy by leveraging publicly available models and data.

In this paper, we propose an ensemble learning approach combining different deep learning macromodels and incorporating effective local features from previous studies. Using a simple ridge regression [59], this approach allows for the prediction of protein functional levels from sequences. We have conducted multiple experiments to demonstrate the effectiveness of this approach, in particular the findings in the ablation experiments of the ensemble and non-ensemble methods. This intriguing finding supports our hypothesis regarding variations in the extracted features across different models. Such discrepancies are likely influenced by the specific protein types and their associated functionalities. Furthermore, the ensemble method not only preserves but also enhances the prediction results obtained merely by using each representation individually across all protein datasets. In fact, on certain protein datasets, the ensemble method exhibits even better performance. This observation suggests that the integrated approach not only retains the feature information obtained from both models but also encompasses a certain level of feature complementarity. The utilization of models trained on distinct datasets with varying architectures is beneficial as it allows diverse features to be captured. The ensemble of multiple features can complement each other, thereby enhancing the prediction of protein mutation effects.

Notably, the presented integration approach is not limited to the models employed in this study, but it can be extended to other pre-trained models. Researchers can integrate various models based on their own understanding in order to enhance predictions for their specific protein species of interest. It is of note that most models used for protein fitness prediction are influenced by multiple sequence alignment (MSA) characteristics and properties of deep mutational scanning (DMS) data, including the number and distribution of conserved regions among homologous sequences. The integration approach can partially compensate for the limited generalizability of models arising from these effects.

In the process of utilizing artificial intelligence for protein engineering tasks, a common scenario is the scarcity of data, making it essential to investigate the influence of data quantity on model prediction. This matter has been thoroughly examined in the research conducted by Barbero et al. [60]. In most cases, models trained on abundant data tend to achieve superior prediction results, as observed in the findings of this study. However, in the case of certain small-scale datasets (TPOR [54] in the S17 less than 300), ensemble methods have demonstrated the ability to enhance prediction accuracy. This advantage stems from the feature information captured by ensemble methods from various large protein databases.

Although the proposed approach in this study generally brings about favorable effects, there are instances where undesired performance can be observed. For instance, when testing on the A0A2Z5U3Z0 [55] dataset within S17, it was found that the ensemble method did not even outperform the unsupervised model in terms of accuracy, although such cases are extremely rare. Potential reasons for this phenomenon include the intricate relationship between measured protein attributes and protein fitness within the data, as well as the inherent inaccuracies resulting from experimental errors during the characterization process. Further investigation and analysis are required to unravel the specific causes. Additionally, this study mainly focuses on the integration of protein-language-based models; future studies will likely embrace multimodal [61] deep learning approaches, where the integrated approach proposed in this paper is expected to play a crucial role in dealing with multimodal information.

## 4. Materials and Methods

### 4.1. Datasets

The training and testing datasets utilized in this study were sourced from ProteinGym [32], a comprehensive collection of deep mutation scanning (DMS) assay datasets encompassing diverse protein types. These datasets are divided into two benchmarks: the substitution benchmark and the indel benchmark. The substitution benchmark comprises high-throughput screening data obtained after carrying out saturating mutation on specific domains or complete sequences of target proteins. This benchmark consists of 87 types, with approximately 1.5 million measured variants. In addition, the indel benchmark involves mutation and screening experimental data acquired before and after amino acid insertion or deletion at specific sites of the target proteins. This benchmark encompasses seven types, with nearly 300,000 measured variants. ProteinGym dataset encompasses a wide variety of protein functional properties, which includes but not limited to thermostability, ligand binding, and aggregation. In addition, it also covers a diverse range of protein families, including kinases, ion channel proteins, and G-protein coupled receptors. The comprehensive nature of these datasets facilitates the exploration of model generalization and allows for the examination of the impacts of different data factors on model performance. In the future, an increasing number of research studies will employ ProteinGym as part of their baseline experimental datasets. This also solidifies the importance of utilizing ProteinGym in this study.

After careful consideration of computational and time costs, we selected a portion of the substitution benchmark as the primary experimental dataset. Although ProteinGym includes measurement data for indel mutations, its quantity is less than a tenth of the substitution data. Moreover, indel mutations are often more complex than substitution mutations. Therefore, investigating complex systems with limited data may not provide a sufficiently objective representation of reality. To capture a more comprehensive representation of the complete dataset, we subset the substitution benchmark based on the following criteria:Inclusion of all taxa present in the full dataset (e.g., humans, other eukaryotes, prokaryotes, viruses);Even distribution of protein sequence lengths (100aa–1000aa);A wide range of sample sizes (ranging from several hundred to several thousand).

With these criteria, we selected 17 DMS datasets, referred to as S17 (Substitution 17) in this study. It can be expected that S17 will accurately reflect the distribution of the complete dataset while minimizing computational and time costs. This subset will facilitate a comprehensive comparison of differences between different models. Figure 4 presents the overview of the complete distribution of S17.

### 4.2. Pre-Training Protein Language Models

When addressing highly complex problems, it is often of great necessity to employ larger, more parameter-rich models to effectively capture the desired information and knowledge [62]. However, using such large models comes with the drawback of extensive tuning costs [63]. In such cases, an alternative strategy can prove to be effective. For a specific class of research objects, one can first train a large pre-trained model on a diverse range of unlabeled samples [64]. This pre-trained model is designed to capture general information and knowledge. Then, a smaller task-specific model is trained separately on different downstream tasks, utilizing the output of the pre-trained model as input features. The main objective of the smaller model is to transform the input into a representation enriched with feature information. This representation is obtained from the pre-trained model’s output and can be used as input for the downstream tasks. Relative to fine-tuning, which typically modifies parameters in the pre-trained model to adapt to new knowledge based on the data characteristics of the downstream task, the above-mentioned strategy does not adjust any parameters in the pre-trained model. One advantage of this approach refers to its ability to utilize the already trained model on any problem falling under the same category. This not only saves computational costs but also facilitates the dissemination and application of the model. This strategy has proven to be particularly common and fruitful in the field of protein engineering within the realm of artificial intelligence [13,58].

Pre-trained protein language models are trained using extensive protein sequence databases, and different models are obtained by training on different datasets, with the extensively utilized UniRef [22,23] series and Pfam [24,25] being prominent examples. The UniRef series comprises a collection of different sequence-clustering databases provided by UniProt, namely UniRef100, UniRef90, and UniRef50, classified according to the varying levels of clustering resolution. Among them, UniRef100 encompasses the largest number of protein sequences. In addition, Pfam serves as a protein family database. Different from the clustering and de-redundancy focus of the UniRef series, Pfam emphasizes protein domains and families, also providing valuable information concerning protein function and structure.

In this study, we primarily concentrated on two pre-trained protein language models: Tranception L [32] and Tape-transformer [58]; they have completed training on different protein datasets. Tranception L trained on the UniRef100 dataset, and the BERT-base [65] model from Tape [58] trained on the Pfam database.

#### 4.2.1. Tranception

Tranception L stands out as the protein language model with the largest parameter count, which boasts approximately 700M parameters, making it the most expansive model observed in this study. Its architecture encompasses 20 attention heads [56] and a total of 36 layers, incorporating several specific optimizations for the prediction of protein fitness. Apparently, it employs Grouped ALiBi position encoding [66], enabling prediction of protein fitness across various types and with any multiple sequence alignment (MSA) depth. Among the tested unsupervised methods, Tranception L emerges as the top overall performer, solidifying its inclusion within the integration framework of this paper.

#### 4.2.2. Tape-Transformer

Tape, a study that evaluates protein embedding tasks, encompasses numerous popular deep learning models, including BERT-base (a transformer model), LSTM [67], and ResNet [68]. These models were trained and tested on diverse downstream tasks, including Fluorescence Landscape Prediction and Stability Landscape Prediction [58], which are highly associated with protein engineering tasks. Transformer-based models, including BERT-base, exhibited superior or equivalent performance compared to other methods. Different from Tranception, which depends on the UniRef100 dataset, Tape-transformer is trained on Pfam.

#### 4.2.3. Details of Model Modifications

While no modifications were applied to the Tape-transformer model, some adjustments were implemented for Tranception L. As Tranception L is originally an unsupervised model designed for the prediction of protein mutation effects, we excluded the inference part of the model. Instead, we retained the output of the model from the “hidden states” component, which served as our feature extractor, converting protein sequences into representations.

Regarding hyperparameter settings, we utilized the default hyperparameters for the Tape-transformer model. In addition, for Tranception L, as each amino acid character is encoded as a 1280-dimensional vector, and with the model boasting over 700M parameters, specific adjustments were essential for efficient execution on a single NVIDIA 3090 GPU. We experimented with different “batch size” values to accommodate protein sequences of varying lengths. Ultimately, we found that a batch size of 16 was suitable for protein sequences of all lengths. Moreover, we set the “scoring window” to the “optimal” option, accommodating longer sequence lengths.

### 4.3. Ensemble Methods for Predicting

To harness the diverse capabilities of various protein language models and effectively model protein sequences in relation to their functions, an ensemble learning approach is employed in this work (Figure 5b). Initially, this approach encodes the features of each input protein sequence based on a pre-trained protein language model. Afterwards, these distinctive features are integrated via feature mapping followed by summation, causing an ensemble representation. Then, this ensemble representation is used to build a regression model and trained with corresponding performance metrics. This is completed to capture the intricate relationship between protein mutants and their mutation effects. Concretely, during the training and testing process, each protein sequence from the deep mutation scanning dataset is subjected to feature encoding, which transforms it into a two-dimensional matrix, also known as “Global Features” [13]. These global features comprise feature vectors encoded for each amino acid letter within the protein sequence. By embodying dense information, these vectors contain specific biochemical contextual knowledge about the respective amino acid’s position in the sequence. This knowledge has been previously acquired by pre-training protein language models on large protein databases.

In order to ensure compatibility with subsequent supervised models, the problem of different dimensions of the feature vectors generated for each amino acid letter in different pre-trained models has to be considered and solved. This entails operations such as feature dimensionality reduction or feature concatenation, enabling the creation of a matrix composed of feature vectors with varying dimensions. Using this approach, we preserve the sequence representation produced by each model while retaining the complete feature information of each amino acid in the sequence. Furthermore, based on the strategy employed in ECNet, we leverage CCMPred (Section 4.4) to predict coupled features for each amino acid in the sequence. This procedure generates a one-dimensional vector, referred to as a “local feature”, for each sequence. Then, the “global features” and “local features” obtained are concatenated for each protein sequence, resulting in a comprehensive “sequence representation” that is rich in feature information. This final representation, along with the corresponding tags, is subsequently fed into the subsequent supervised model for training. In the context of the deep mutation scanning dataset, considering the distinct quantitative characteristics, separate modeling is performed for each specific protein and its corresponding mutant sequences. Moreover, this ensures the establishment of a mapping relationship between the sequences and their specific functional quantification for the current protein data.

### 4.4. CCMPred and Local Representation

CCMPred [38] is an algorithm that is capable of computationally identifying coevolutionary indices between residue pairs in proteins. It operates efficiently on GPUs. The underlying principle of CCMPred capitalizes on the conservation of evolutionary sequences. It utilizes a statistical model based on Markov random fields [69] to learn evolutionary couplings in multiple sequence alignments (MSA) of target protein homologs. That is to say, it estimates the mutual dependencies of all pairs of residues based on the observed MSA. The computed couplings have been demonstrated in multiple studies to associate with protein fitness. In this study, similar to ECNet [39], we adopt the approach of using CCMPred-calculated couplings as a local representation of protein sequences. This inclusion is aimed at enhancing the feature recognition capability of deep learning models on a wide range of protein sequences. To achieve the same objective, we follow ECNet’s methodology and incorporate stacked sequences of evolutionary couplings computed by CCMPred into our ensemble model.

## 5. Conclusions

To conclude, by employing a multi-model integration strategy, we present a novel approach for protein mutation effect prediction in this study. This approach leverages the unique architectural advantages of different models and the synergistic effects of diverse training data, allowing us to unlock the full potential of existing large-scale models. To evaluate the performance of our proposed approach, we conduct comprehensive benchmarking and ablation experiments on the carefully selected deep mutation scanning dataset, S17. Our experimental results demonstrate that the models utilizing the integrated approach consistently outperform both the previously published methods and models discussed in this study, with statistically superior average predictive performance. Moreover, these promising outcomes underscore the potential of our approach to significantly enhance the workflow of directed evolution, providing valuable assistance in the development of novel proteins with improved functional characteristics.

## Figures and Tables

**Figure 1 ijms-24-16496-f001:**
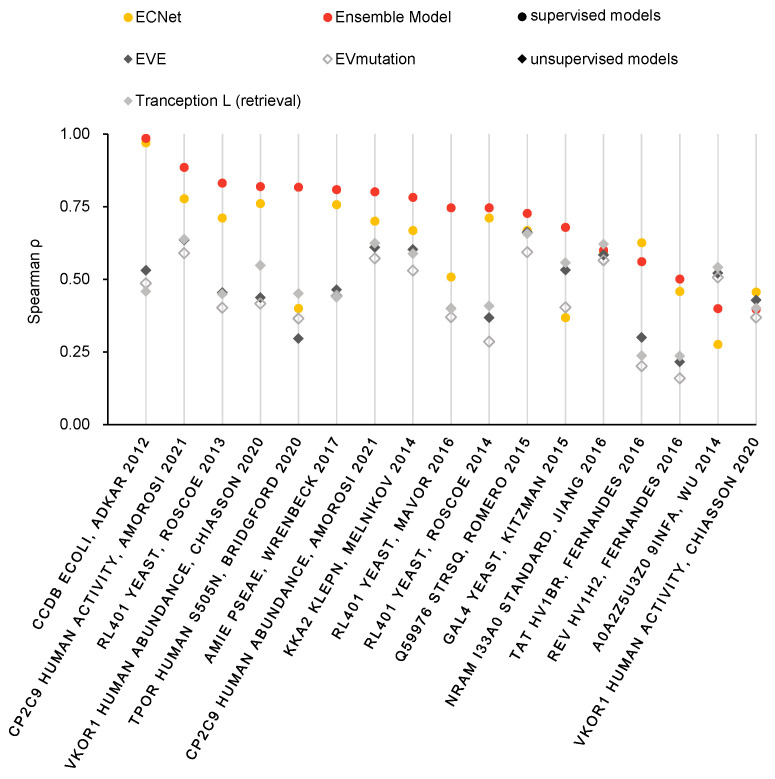
The performance of ensemble model and other unsupervised and supervised models on S17 [42,43,44,45,46,47,48,49,50,51,52,53,54,55]. The ensemble model achieves the best prediction on 15 of the 17 deep mutation scans, and, rightfully, due to the full utilization of the available data, the prediction performance of the supervised model is generally higher than that of the unsupervised model.

**Figure 2 ijms-24-16496-f002:**
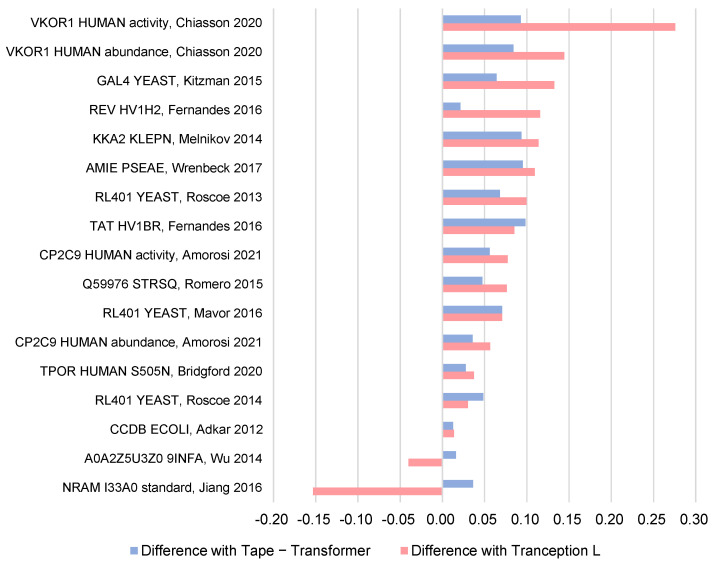
The impact of using ensemble vs. non-ensemble methods on forecasting results. Tranception-ridge and Tape-transformer-ridge are ridge regression models constructed using global features obtained from only one of the pre-trained language models. The results presented in this figure are the difference between the Spearman R obtained by the ensemble model and the prediction obtained by these two models, respectively, on S17 [42,43,44,45,46,47,48,49,50,51,52,53,54,55]. Larger values represent better results for the ensemble model.

**Figure 3 ijms-24-16496-f003:**
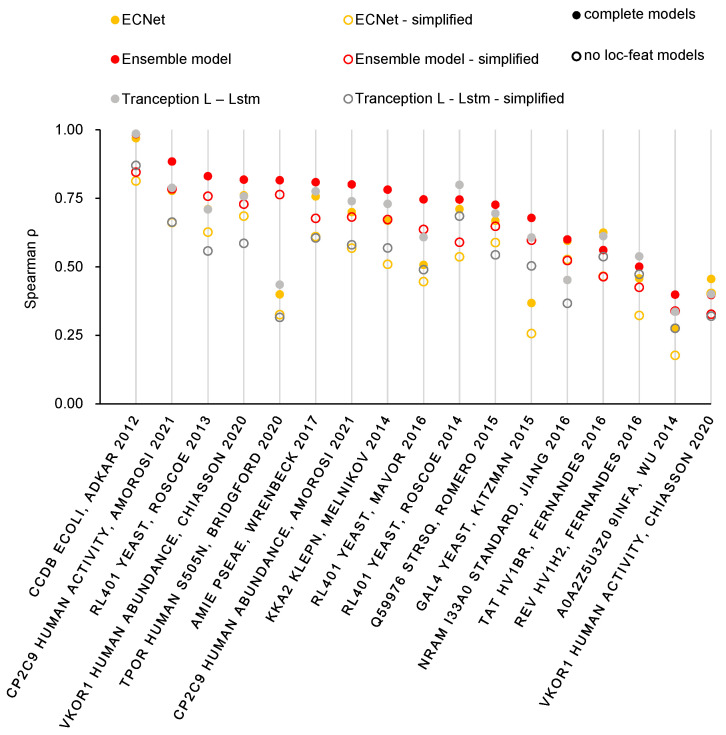
Prediction effects of models with and without “Local feature” on S17 [42,43,44,45,46,47,48,49,50,51,52,53,54,55].

**Figure 4 ijms-24-16496-f004:**
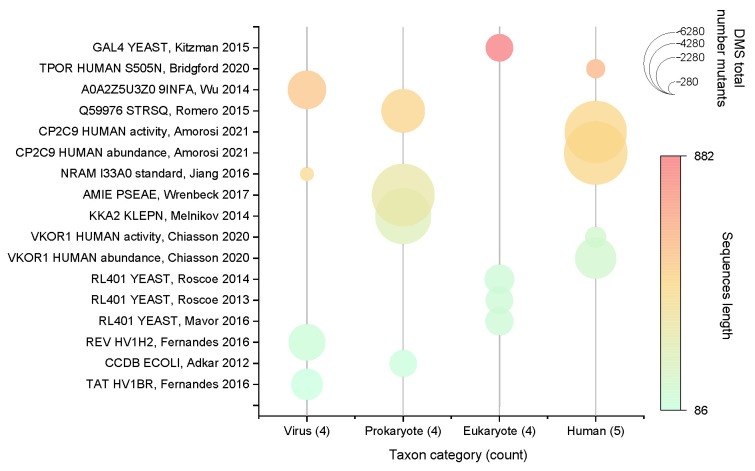
Distribution of protein species and amino acid sequence lengths contained in S17 [42,43,44,45,46,47,48,49,50,51,52,53,54,55]. In the DMS data contained in S17, the maximum number of mutants measured was 6370 and the minimum was only 298. The shortest sequence length of the measured mutants was only 86 amino acids, the longest was 881 amino acids, and the rest of the lengths were evenly distributed in between. A total of four categories of species are included in S17, and the number of these four categories is almost equal.

**Figure 5 ijms-24-16496-f005:**
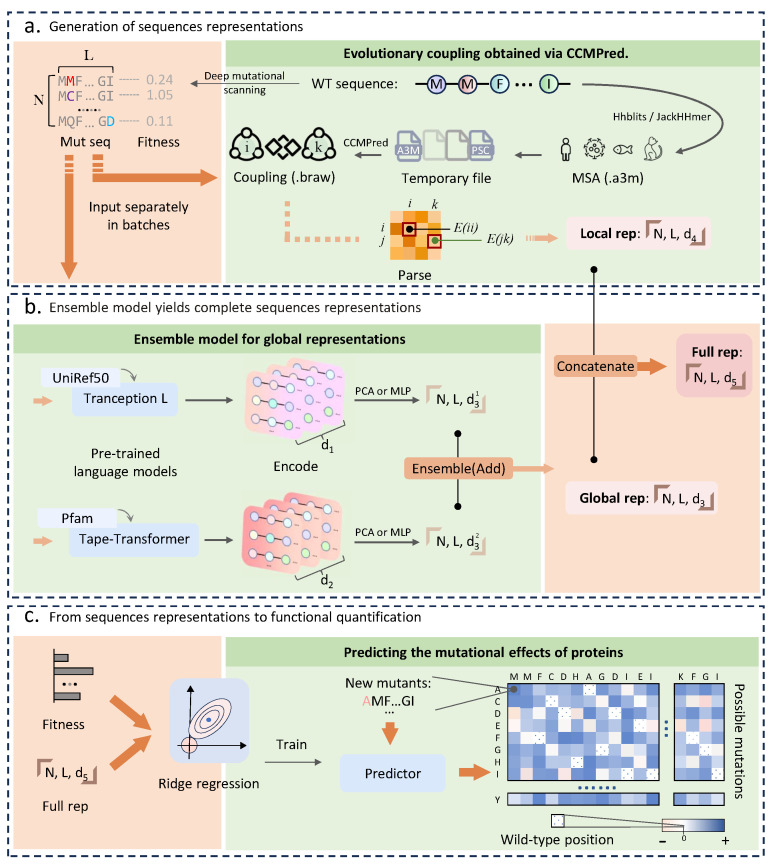
Ensemble model workflow for the protein mutation effect prediction. Each protein sequence is encoded into a learnable sequence representation consisting of local and global representations concatenated. (**a**) For the local sequence representations, the WT sequence of the mutants of interest is input into CCMPred by MSA to obtain the coupling relationships between residues on this protein sequence, and then the local representations of each mutant sequence are resolved by this coupling relationship for concatenation. (**b**) The global representations of sequences are obtained by integrating the outputs of different pre-trained language models in an ensemble model, which is then concatenated with the local representations to make the full representations of the sequences for training and prediction. (**c**) The complete sequence representation and the measured fitness corresponding to it are taken as inputs, and the mapping relationship between sequence representation and fitness is obtained by a ridge regression algorithm. After training, the model can predict the mutational effects of new protein mutants of interest.

## Data Availability

The Tranception and ProteinGym data are available at https://github.com/oatml-markslab/tranception, (accessed on 20 July 2023). The source code of the ECNet can be found in the GitHub repository: https://github.com/luoyunan/ECNet#generate-local-features-using-hhblits-and-ccmpred, (accessed on 2 June 2022). A tutorial on how to use CCMpred can be found at https://github.com/soedinglab/CCMpred, (accessed on 27 August 2020).

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
