# Peer review of "Ensemble Learning with Supervised Methods Based on Large-Scale Protein Language Models for Protein Mutation Effects Prediction"

_ijms, 2023, doi:10.3390/ijms242216496_

Round 1
Reviewer 1 Report
Comments and Suggestions for Authors
The authors present an analysis of ensemble learning applied to the prediction of mutation effects on proteins.
The manuscript has a major problem at the moment, which is that it is not written in a way that is helpful to the reader. A good manuscript provides the information that the reader needs by way of background in the Introduction. Then, the Results are presented without commentary, in a neutral way for the reader to review. Finally, the results are discussed and put into context in Discussion. The Method and materials are described separately. At the moment, the manuscript is very jumbled and difficult for me to read.
Specifically, the results section contains a large amount of introduction text that does not relate at all to the authors' actual results. For example, Lines 148 to 165 almost entirely discuss the authors methods, and do not contain any results. Why is this section not in methods? Figure 1 is a method visualisation, I cannot see any results in it, why is it not in methods? Similarly, line 266 starts to discuss the results, describing them as compelling. Why is this text discussing the results not in discussion? The results should be presented in a factual way, not mixed up with the authors' opinions as to whether they are compelling or not.
In my view the scientific content of the manuscript is interesting and worthy of publication, but the structure of the manuscript makes it difficult to access.
Comments on the Quality of English LanguageThe standard of writing is good.
Reviewer 2 Report
Comments and Suggestions for Authors
This manuscript presents a notable contribution to the field of protein engineering with an ensemble learning approach that integrates multiple protein language models. The method's potential to enhance the quantitative prediction of mutational effects and accelerate molecular design is praiseworthy. The use of 17 deep mutation scanning datasets to validate this approach and demonstrate improved prediction accuracy over traditional methods is particularly commendable.
Upon thorough review, it is evident that this manuscript would benefit from revisions to enhance the clarity and professionalism of the English language employed. Specific spelling errors have been identified and should be rectified:
Corrections:
In the abstract, line 27, "information" is incorrectly spelled as "inforamtion".
In the introduction, line 33, the word "practically" is misspelled as "practicaily".
In the title of section 2.1, "protein" should begin with a capital 'P'.
In section 2.3, line 407, the word "size" in "batch size" is incorrectly spelled as "sieze".
The title of section 4.1, currently "Datasts," should be corrected to "Datasets".
In section 4.3, line 401, "output" is incorrectly spelled as "outptut".
On line 122, the word "approaches" is misspelled as "approcahes".
On line 163, "employed" is incorrectly spelled as "epmloyed".
On line 248, "pertains" is misspelled as "perteins".
On line 283, there is an indicated misspelling of "encompasses"; the correct spelling should be provided.
On line 355, "specific" is misspelled as "specifi".
On line 457, "modifications" is incorrectly spelled as "modificatons".
On line 489, "advantage" is misspelled as "advantaeg".
In addition, it is recommended to minimize the use of redundant expressions and shorten lengthy sentences to improve readability and logical flow for the reader.
Lastly, incorporating a discussion of the study's limitations would enhance the manuscript's objectivity and comprehensiveness. For example, addressing how the relatively small size of the S17 dataset might impact the results could offer a more balanced perspective on the research findings.
Comments on the Quality of English LanguageThere are sections where the English expression is redundant, making the text difficult to read and obscuring the understanding of the research content and the argument.
Round 2
Reviewer 1 Report
Comments and Suggestions for Authors
Dear authors,
Thank you for improving the flow of the manuscript. There are some very minor typographical errors, examples mentioned below. Aside from these, in my view the paper is worthy of publication and should be accepted.
Line 341, "scenarios(Figure 5)," is missing a space
Line 344, "Averagely, we observed" should be "On average, we observed
Line 354, ""large models"[13].In" is missing a space
Comments on the Quality of English Language
In general the manuscript is written to a good standard.